# Occurrence of Antimicrobial-Resistant *Escherichia coli* in Marine Mammals of the North and Baltic Seas: Sentinels for Human Health

**DOI:** 10.3390/antibiotics11091248

**Published:** 2022-09-14

**Authors:** Stephanie Gross, Anja Müller, Diana Seinige, Peter Wohlsein, Manuela Oliveira, Dieter Steinhagen, Corinna Kehrenberg, Ursula Siebert

**Affiliations:** 1Institute for Terrestrial and Aquatic Wildlife Research, University of Veterinary Medicine Hannover, Foundation, Werftstraße 6, 25761 Büsum, Germany; 2Institute for Veterinary Food Science, Justus Liebig University Giessen, Frankfurter Str. 92, 35392 Giessen, Germany; 3Office for Veterinary Affairs and Consumer Protection, Ministry of Lower Saxony for Food, Agriculture and Consumer Protection, Alte Grenze 7, 29221 Celle, Germany; 4Department of Pathology, University of Veterinary Medicine Hannover, Bünteweg 17, 30559 Hannover, Germany; 5CIISA—Centre for Interdisciplinary Research in Animal Health, Faculty of Veterinary Medicine, University of Lisbon, Avenida da Universidade Técnica, 1300-477 Lisbon, Portugal; 6Associate Laboratory for Animal and Veterinary Sciences (AL4AnimalS), Avenida da Universidade Técnica, 1300-477 Lisbon, Portugal; 7Fish Disease Research Unit, University of Veterinary Medicine Hannover, Bünteweg 17, 30559 Hannover, Germany

**Keywords:** pinnipeds, cetaceans, wildlife, Enterobacteriaceae, multidrug resistance, One Health

## Abstract

Antimicrobial resistance is a global health threat that involves complex, opaque transmission processes in the environment. In particular, wildlife appears to function as a reservoir and vector for antimicrobial-resistant bacteria as well as resistance genes. In the present study, the occurrence of antimicrobial-resistant *Escherichia coli* was determined in marine mammals and various fish species of the North and Baltic Seas. Rectal or faecal swabs were collected from 66 live-caught or stranded marine mammals and 40 fish specimens. The antimicrobial resistance phenotypes and genotypes of isolated *E. coli* were determined using disk diffusion tests and PCR assays. Furthermore, isolates were assigned to the four major phylogenetic groups of *E. coli*. Additionally, post mortem examinations were performed on 41 of the sampled marine mammals. The investigations revealed resistant *E. coli* in 39.4% of the marine mammal samples, while no resistant isolates were obtained from any of the fish samples. The obtained isolates most frequently exhibited resistance against aminoglycosides, followed by β-lactams. Of the isolates, 37.2% showed multidrug resistance. Harbour porpoises (*Phocoena phocoena*) mainly carried *E. coli* isolates belonging to the phylogenetic group B1, while seal isolates were most frequently assigned to group B2. Regarding antimicrobial resistance, no significant differences were seen between the two sampling areas or different health parameters, but multidrug-resistant isolates were more frequent in harbour porpoises than in the sampled seals. The presented results provide information on the distribution of antimicrobial-resistant bacteria in the North and Baltic Seas, and highlight the role of these resident marine mammal species as sentinels from a One Health perspective.

## 1. Introduction

Antimicrobial-resistant bacteria increasingly pose a global challenge for the successful treatment of bacterial infections in both humans and animals [1,2,3]. Besides the development of new antimicrobial drugs [4], existing antimicrobials need to be used appropriately, as their usage drives the development of resistant strains via positive selection pressure [5,6,7]. After the emergence of resistant strains, genetic material inducing resistance can be transferred within bacterial populations as well as between different bacterial species [8]. Nowadays, considerable amounts of antimicrobial-resistant bacteria can be found not only in humans, companion animals and livestock [9,10], but also in wildlife [11,12,13,14,15] and the environment [16,17,18,19]. Although antibiotics and antimicrobial resistance are known to occur naturally in the environment [16,20,21], different anthropogenic sources have been identified as additional sources of environmental contamination [22,23,24,25,26]. Wildlife most probably acquires antimicrobial-resistant bacteria from environmental sources [11,27,28], and can also act as a reservoir and vector for infection or colonisation [15,29,30,31]. In this context, individual wildlife species could be used as effective bioindicators to display the level of environmental contamination as well as the potential of a public health risk caused by antimicrobial-resistant bacteria.

Marine mammals have been postulated as being sentinels of marine ecosystem health, especially due to their long life spans and their role as apex predators [32,33,34]. Moreover, they are prime barometers for human health (as humans share food and coastal habitats with marine mammals), and can thus indicate potential negative impacts like environmental contamination at an early stage [34]. Aquatic ecosystems are prone to high levels of anthropogenic impact [35] with regard to the release of antimicrobial-resistant bacteria, especially via sewage discharges, agricultural run-offs and health-care effluents [19,22,23,24,25,26,36,37]. Once these microbes and antimicrobial residues are released into the environment, natural water bodies themselves may play a certain role in the persistence, emergence, and spread of antimicrobial resistance [38]. However, antimicrobial-resistant bacteria have also been found in various marine species of all taxa [39,40,41,42,43,44,45]. For surveillance, apex predators like marine mammals are of specific interest, as they may mirror the burden of antimicrobial resistance in the marine food web as well as the environment.

*Escherichia coli* are globally established indicators of anthropogenic faecal contamination as well as the pollution of waterways and coastal ecosystems [46,47,48]. Furthermore, *E. coli* isolated from surface waters have been identified as important reservoirs of antimicrobial-resistant bacteria and resistance genes [49]. In addition, *E. coli* can easily receive, harbour and transfer antimicrobial resistance genes, mainly via horizontal gene transfer to other bacteria [50,51]. In aquatic environments, *E. coli* can survive for varying periods [52,53]. As commensals of the intestines of mammals and birds, *E. coli* are widespread, with some pathogenic strains causing severe intestinal or extra-intestinal diseases [54]. Furthermore, similar *E. coli* strains are shared by humans, wildlife and the environment [55], indicating a potential hazard to public health [36,56]. Taken together, these facts highlight *E. coli* as an effective indicator bacterium for antimicrobial resistance in aquatic environments.

The present study aimed to determine the occurrence of antimicrobial-resistant *E. coli* in four different marine mammal species of the North and Baltic Seas. Various fish species were included to investigate the potential colonisation of the marine mammals via their food web. Differences in the burden of antimicrobial resistance in North and Baltic Seas were evaluated using marine mammals as aquatic sentinel species. Additionally, the level of multidrug resistance among the isolates was determined, as these isolates pose a potential public health risk from a One Health perspective.

## 2. Material and Methods

### 2.1. Study Species

Samples in the present study were obtained from the three resident marine mammal species of the German North and Baltic Seas, the harbour porpoise (*Phocoena phocoena*), the harbour seal (*Phoca vitulina*), and the grey seal (*Halichoerus grypus*), as well as one resident species of the northernmost part of the Bothnian Bay, Baltic Sea, the ringed seal (*Pusa hispida*). The harbour porpoises belong to two different populations, namely the North Sea population and the Western Baltic Sea population [57,58,59]. While there is only little overlap between these two populations, the Western Baltic Sea population has a certain geographic overlap with the Baltic Proper population [60]. The harbour seals, as well as the grey seals of the North and Baltic Seas, can also been seen as different stocks, with only little interchange between the seals of the different seas [61,62,63,64]. The ringed seal inhabits the Arctic and sub-Arctic regions, including the northern region of the Baltic Sea [65]. It only rarely occurs as a visitor in the German North or Baltic Seas. In total, 66 samples of marine mammals were included in the present study, 30 from the North Sea and 36 from the Baltic Sea, including 16 harbour porpoises (two from the North Sea, 14 from the Baltic Sea; four neonates, seven juveniles, five adults), 24 harbour seals (21 from the North Sea, three from the Baltic Sea; one neonate, seven juveniles, 16 adults), 10 grey seals (seven from the North Sea, three from the Baltic Sea; two juveniles, one adult, seven unclassified), and 16 ringed seals (all from the Baltic Sea, 14 from the Bothnian Bay, two from the coast of Schleswig-Holstein; nine juveniles, seven adults).

Furthermore, different fishes were sampled from the North and Baltic Seas. In total, 40 samples of 12 different fish species were obtained, 20 each from the North Sea and the Baltic Sea. Sampled fishes included three Atlantic herrings (*Clupea harengus*) (one from the North Sea, two from the Baltic Sea), seven plaices (*Pleuronectes platessa*) (two from the North Sea, five from the Baltic Sea), three Atlantic cods (*Gadus morhua*) (all from the Baltic Sea), nine common dabs (*Limanda limanda*) (two from the North Sea and seven from the Baltic Sea), three European flounders (*Platichthys flesus*) (all from the Baltic Sea), six Atlantic mackerels (*Scomber scombrus*) (all from the North Sea), four European smelts (*Osmerus eperlanus*) (all from the North Sea), and from the North Sea single individuals of solenette (*Buglossidium luteum*), European sea sturgeon (*Acipenser sturio*), Grey gurnard (*Eutrigla gurnardus*), European pilchard (*Sardina pilchardus*), and whiting (*Merlangius merlangus*).

### 2.2. Study Area

The present study investigated samples of marine mammals and fishes of two different seas, the North Sea and the Baltic Sea. Both are connected via the Skagerrak, Kattegat, and Danish straits. While the North Sea is a marginal sea of the eastern Atlantic Ocean, the Baltic Sea is a brackish inland sea of the Atlantic. The North Sea has a high water exchange rate with the Atlantic Ocean, to which it is connected via the English Channel in the South and the Norwegian Sea in the North. At the German North Sea coast, there is a usual tidal range of around two to three meters. In comparison to this, the Baltic Sea has a low water exchange rate with the Atlantic Ocean, as it is almost completely surrounded by dry land, with only the above-mentioned natural connection via the North Sea to the Atlantic Ocean. The tidal range in the Western Baltic Sea is usually around 30 cm, and decreases to zero moving eastwards. Both Seas are densely populated by humans and are subject to highly diverse anthropogenic impacts.

### 2.3. Sample Collection

Samples of marine mammals were obtained either as swabs of fresh faeces from the ground or as rectal swabs of alive or dead animals (Sterile transport swabs, Heinz Herenz, Germany). Used swabs were directly stored in a transport gel medium. Faecal samples were taken at haul-out sites of seals by inserting the sterile, dry swab into the excrement and turning it a few times to gain enough material. Rectal swabs of live seals were taken during regular seal catches for health assessments, while two live harbour porpoises were accidentally caught in a fishing net and rectal swabs were obtained before release. Furthermore, rectal swabs were taken of stranded dead marine mammals that showed a decomposition status of three or better, with the exception of two animals with decomposition status four. Dead marine mammals were either stranded deceased or were mercy killed on the beach by authorized and specially trained members of the marine mammal stranding network of Schleswig-Holstein. Ringed seals in Sweden were legally hunted in accordance with the hunting quota by Swedish hunters. For rectal samples, sterile swabs were inserted into the rectum and subsequently turned several times on the mucosa. All marine mammal swabs were taken between 30 March 2017 and 8 June 2019. 

Fish samples were partly taken during research surveys carried out by the Johann Heinrich von Thünen Institute in both the North and Baltic Seas (sample ID 74–86, 100–106). The sampling in the Baltic Sea was performed in March 2018 during the cruise Solea SB746, BITS (Baltic International Trawl Survey), the one in the North Sea took place in June 2019 during the cruise Solea SB764 HERAS (the 2019 ICES Coordinated Acoustic Survey in the Skagerrak and Kattegat, North Sea, West of Scotland and the Malin Shelf area). All other fishes were obtained from local fishermen or private anglers. Samples were gained as rectal swabs or, in the case of fish that were too small, as intestinal swabs. Fish species were identified on the basis of morphological parameters [66]. Fish samples were taken between 26 November 2017 and 15 July 2019.

A map created with ArcGIS (ESRI, Version 10.6.1) indicates all sample locations for the marine mammals as well as for the fishes (Figure 1). A more detailed map for the samples of the North and Baltic Sea coasts of Schleswig-Holstein can be found in the Appendix A. Information on all samples, including species, sample area, sampling date and method, age class, and sex of sampled animals, as well as if a necropsy was performed, are listed in the Appendix A.

### 2.4. Necropsies

The stranding network of the German federal state of Schleswig-Holstein retrieves stranded marine mammals off the coast of Schleswig-Holstein, which then undergo post mortem examinations at the Institute for Terrestrial and Aquatic Wildlife Research (ITAW), University of Veterinary Medicine Hannover, Foundation, Büsum, Germany [67,68,69]. In the present study, 41 of the 66 sampled marine mammals underwent post mortem examinations, namely nine harbour seals (six from the North Sea, three from the Baltic Sea), two grey seals (one from the North Sea, one from the Baltic Sea), 16 ringed seals (all from the Baltic Sea) and 14 harbour porpoises (two from the North Sea, 12 from the Baltic Sea). Of these 41 animals, 17 were stored frozen after sampling and prior to necropsy, all others were necropsied directly after recovery. The decomposition states varied from fresh, through good, to moderate, except of two animals with poor decomposition status (sample ID 1 and 148).

Pathological investigations were performed in accordance with a standardised protocol [70,71]. All carcasses underwent a full post mortem investigation. Harbour porpoises were classified into the following age classes depending on their total length: neonate (<1.0 m), juvenile (1.0–1.3 m) and adult (>1.3 m). Seals were classified in accordance with their habitus, their stranding date, and the known pupping season into the three classes neonate, juvenile and adult. The nutritional status of the carcasses was judged depending on the blubber thickness and muscle development [70]. Furthermore, samples were taken for histopathological, parasitological, and microbiological investigations. These samples were taken and investigated as previously described [70]. The extent of histopathological investigations depended on the decomposition status and macroscopical findings. Histopathological samples were obtained from at least the main organs (lung, liver, spleen, kidney, and intestine) in all but four carcasses (sample ID 1, 81, 83, and 85), and additionally from organs and tissues with morphological changes. They were fixed in 10% buffered formalin, and subsequently embedded in a paraffin wax, cut into 3 µm slices, and stained with haematoxylin and eosin. Parasites were classified on the basis of their morphological characteristics. From 38 carcasses, tissues submitted for microbiological testing included at least the lung, liver, spleen, kidney, and intestine in all but two animals (sample ID 1 and 81). 

### 2.5. E. coli Isolation and Identification

Obtained swabs were stored at room temperature for between one week and almost nine months. After initial selective enrichment for Enterobacteriacae in Mossel bouillon, samples were subsequently transferred to selective plates containing antimicrobials as sample processing aimed directly on the isolation of only resistant *E. coli*. Mossel bouillon (Carl Roth GmbH & Co. KG, Karlsruhe, Germany) was blended, and Blood agar (Carl Roth GmbH & Co. KG) and Gassner agar (Sigma Aldrich Chemie GmbH, Steinheim am Albuch, Germany) plates were poured in accordance with the manufacturers’ recommendations. Furthermore, MacConkey agar (Carl Roth GmbH & Co. KG) plates with the addition of eight different antimicrobials were poured in accordance with the manufacturer’s information using the following antibiotics: ampicillin (30 mg/L; Carl Roth GmbH & Co. KG), cephalothin (30 mg/L; TCI Europe N.V., Zwijndrecht, Belgium), chloramphenicol (10 mg/L; Carl Roth GmbH & Co. KG), ciprofloxacin (1 mg/L; Alfa Aeser, Thermo Fischer Scientific, Waltham, MA, USA), colistin (2 mg/L; AppliChem GmbH, Darmstadt, Germany), gentamicin (10 mg/L; Carl Roth GmbH & Co. KG), sulfamethoxazole (512 mg/L; Sigma Aldrich Chemie GmbH), and tetracycline (15 mg/L; AppliChem GmbH). First, each swab was streaked on Blood and Gassner agar before being placed into Mossel bouillon. Agar plates and bouillon were incubated for 24 h at 37 °C. Here, detection of colonies on blood and Gassner agar plates, as well as blurring of the bouillon, were used as references for bacterial growth. Subsequently, 100 µL of the Mossel bouillon was streaked on each of the eight different antimicrobial agar plates, which were then incubated for 24 h at 37°C. From the antimicrobial containing agar plates that showed bacterial growth, five colonies were sampled. In cases where less than five colonies grew on one plate, all colonies were sampled. The selected colonies were then suspended in LB medium (Carl Roth GmbH & Co. KG) and incubated for 24 h at 37 °C. After incubation, 870 µL of the LB medium were compounded with 130 µL glycerine, vortexed and stored at −80 °C. Colonies were selected based on their macroscopic appearance. On MacConkey agar, the typical colour of *E. coli* colonies (among others, e.g., *Klebsiella* and *Enterobacter*) is red or pink with a hazy medium surrounding the colony.

A preselection of potential *E. coli* was performed using selective plating on Chromocult (Merck KGaA, Darmstadt, Germany), Gassner and MacConkey agar. As *E. coli* ferments lactose, the typical appearance of its colonies is blue to violet on Chromocult, blue on Gassner, and red or pink on MacConkey agar. For genomic DNA isolation, presumptive *E. coli* isolates were cultured overnight, suspended in 300 µL bidistilled water, then heated at 99 °C for 15 min and centrifuged (13,000× *g*) for 2 min. Species confirmation of isolates was performed as previously described via *gadA*-PCR [72]. Isolates that did not react in this PCR-assay were analysed via MALDI-TOF for species identification. By this means, 116 *E. coli* isolates were confirmed. As each sample was streaked on eight different plates and up to five colonies per plate were collected, one sample delivered one to five isolates.

### 2.6. Resistance Phenotype

Resistance profiles of the 116 confirmed *E. coli* isolates were determined via disk diffusion tests in accordance with the Clinical and Laboratory Standards Institute (CLSI) standards [73]. Antimicrobial infused disks were purchased from Oxoid (Wesel, Germany) and included the following 14 antimicrobials: amoxicillin/clavulanic acid (20/10 µg), ampicillin (10 µg), cefazolin (30 µg), cefpodoxime (10 µg), chloramphenicol (30 µg), florfenicol (30 µg), streptomycin (10 µg), gentamicin (10 µg), kanamycin (30 µg), ciprofloxacin (5 µg), nalidixic acid (30 µg), compound sulphonamide (300 µg), trimethoprim (5 µg), and tetracycline (30 µg), representing seven different antimicrobial classes. Isolates showing resistance to cefpodoxime were regarded as presumptive extended-spectrum β-lactamase (ESBL)/AmpC-producers and were subjected to confirmatory testing for ESBL production by disk diffusion according to CLSI standards [74]. To achieve this, the inhibition zones for cefotaxime (30 µg) and ceftazidime (30 µg) as single substances were compared to those of cefotaxime and ceftazidime in combination with clavulanic acid (10 µg). An increase of ≥5 mm was considered indicative of ESBL production. If resistance to either one or both of the substances was observed but inhibition zones increased less than 5 mm in the presence of clavulanic acid, the isolates were regarded as presumptive AmpC-producers [74].

For each isolate, a bacterial suspension in NaCl with 0.5 on the MacFarland scale, corresponding to approximately 10^8^ CFU/mL, was prepared and streaked on Mueller–Hinton (Carl Roth GmbH & Co. KG) agar plates. Subsequently, up to six antimicrobial disks were placed on the agar surface and incubated for 18 h at 35 °C ± 2 °C. Inhibition zones were evaluated in accordance with CLSI standards [74,75]. For subsequent analyses, isolates with intermediate susceptibility and resistant isolates were grouped together. Potential colistin-resistant isolates were further tested using MacConkey agar plates supplemented with colistin (2 mg/L), representing an additional antimicrobial class. Copy isolates were avoided by using only one of the isolates that were obtained from the same swab and showed the same resistance profile. 

### 2.7. Resistance Genotype

Isolates were tested for the presence of antimicrobial resistance genes using a panel of PCR assays as described previously [76]. The targeted genes comprised *strA*, *strB* [77], *aadA1* [78], *aadA2* [79], *ant-(2″)-I*, *aac(3)-II* and *aac(3)-IV*, mediating resistance to aminoglycosides; *bla*_TEM_, *bla*_SHV_, *bla*_OXA-1-like_, and *bla*_OXA-2_, mediating resistance to β-lactams; presumptive AmpC-producers were further tested for the presence of the acquired AmpC β-lactamase gene *bla*_CMY_; *aac(6‘)-Ib-cr*, *qnrA*, *qnrB*, *qnrC*, *qnrD* and *qnrS*, associated with reduced susceptibility to quinolones; *catA1*, *catA2*, *catA3*, *catB2*, *catB3*, *cmlA* and *floR*, mediating resistance to phenicols; *sul1*, *sul2* and *sul3*, mediating resistance to sulfonamides; *dfrA1/15/16*, *dfrA5/14*, *dfrA7/17* and *dfrB1/2/3*, mediating resistance to trimethoprim; and *tet*(A), *tet*(B), *tet*(C), *tet*(D), *tet*(E), *tet*(G), *tet*(H), *tet*(L) [80], *tet*(M) and *tet*(O), mediating resistance to tetracyclines [76]. If isolates reacted with primers targeting *bla*_CMY_, *dfrA* or *dfrB* genes, the amplicons were purified using a QIAquick PCR Purification Kit (QIAGEN, Venlo, The Netherlands) and sequenced using the sanger sequencing service provided by Eurofins Genomics (Ebersberg, Germany). The resulting sequences were analysed using the BLASTN algorithm (https://blast.ncbi.nlm.nih.gov/Blast.cgi, accessed on 14 July 2022) in order to determine the specific type of resistance gene.

### 2.8. Molecular Typing

PCR assays were used to assign all *E. coli* isolates to one of the four major phylogenetic groups (A, B1, B2, and D) by targeting two genes (*chuA* and *yjaA*) [81] and the anonymous DNA fragment TSPE4.C2 [72]. Macrorestriction analyses with XbaI digestion according to the PulseNet protocol for *Salmonella*, *Shigella*, *E. coli* O157 and other Shiga toxin-producing *E. coli* [82] was used to investigate the genetic relatedness of the isolates. DNA fragments were separated by pulsed-field gel electrophoresis using a CHEF DR II system (BioRad, Hercules, CA, USA) with the following settings: 6 V, initial time 6.8 s, final time 35.4 s, 20 h run time. *Salmonella* Typhimurium strain LT2 served as a size marker. A dendrogram depicting band pattern similarity was created based on the Unweighted Pair Group Method with Arithmetic Mean (UPGMA) using BioNumerics software version 7.6 (Applied Maths, Sint-Martens-Latem, Belgium) and applying the Dice coefficient with 1% position tolerance and 0.5% optimization. Band patterns are shown in the Appendix A.

### 2.9. Statistical Analysis

RStudio version 1.4.1103 (R version 4.1.2) with the packages readxl and DescTools was used to perform statistical analyses. The significance of the association between (i) the number of samples with resistant *E. coli* and (ii) the number of samples with multidrug-resistant *E. coli* and the variables area (North and Baltic Sea) and marine mammal species was tested via Fisher’s exact test because of the small sample size. Furthermore, for necropsied animals, the number of samples with resistant *E. coli* was tested against the variables overall health status, body condition, age, sex, sepsis, and (broncho)pneumonia. The level of significance was set at *p* < 0.05.

## 3. Results

### 3.1. Occurrence of Antimicrobial-Resistant E. coli Isolates

For the present study, 66 samples were collected from marine mammals and 40 from different fish species. Of the marine mammal samples, 58 (87.9%) showed microbial growth on at least one of the antimicrobial-containing agar plates, as also observed for 16 (40%) of the fish samples. At least one antimicrobial-resistant *E. coli* isolate was obtained from 26 (39.4%) of the marine mammal samples, whereas the identity of the other collected isolates could not be confirmed as *E. coli*. No resistant *E. coli* was isolated from any of the fish samples. Resistant *E. coli* was obtained from 16 (44.4%) of the 36 marine mammal samples from the Baltic Sea as well as from 10 (33.3%) of the 30 marine mammal samples from the North Sea. Of the different species, the resistant isolates were most frequent in ringed seals, with 10 (all Baltic Sea) of 16 samples carrying resistant *E. coli* isolates. Nine (three Baltic Sea, seven North Sea) of 24 harbour seal samples, three (all North Sea) of 10 grey seal samples and four (three Baltic Sea, one North Sea) of 16 harbour porpoises’ samples carried resistant *E. coli* isolates. The numbers of samples with resistant *E. coli* across all marine mammals as well as separated by species and seas are depicted in Figure 2. No significant differences were detected in the occurrence of resistant *E. coli* either between the two areas (*p* = 0.4504) or between the different species (*p* = 0.1602).

Of the 116 *E. coli* isolates confirmed originally, 63 were identified as duplicates. Ten isolates did not show any resistance in the disk diffusion test. In total, 43 *E. coli* isolates with phenotypic (intermediate) resistances were obtained from the 66 marine mammal samples (Figure 3), while no isolate from the fish samples could be confirmed as *E. coli*. The 43 isolates were distributed among the species and sampling areas as follows: 10 isolates from four harbour porpoises (one isolate from the North Sea, nine from the Baltic Sea); 13 isolates from nine harbour seals (eight isolates from the North Sea, five from the Baltic Sea); three isolates from three grey seals (all from the North Sea); and 17 isolates from 10 ringed seals (all from the Baltic Sea).

### 3.2. Resistance Phenotypes and Genotypes

Overall, the results from the disk diffusion tests showed that the 43 *E. coli* isolates had the highest resistance rate against streptomycin (28/43, 65.1%), followed by resistances to cefazolin (22/43, 51.2%) and ampicillin (21/43, 48.8%). Resistance was also frequently seen against amoxicillin/clavulanic acid (14/43, 32.6%), tetracycline (14/43, 32.6%) and sulfonamides (13/43, 30.2%), followed by nalidixic acid (10/43, 23.3%), ciprofloxacin (9/43, 20.9%), chloramphenicol (7/43, 16.3%) and trimethoprim (7/43, 16.3%). Kanamycin (6/43, 14%) and florfenicol (5/43, 11.6%) followed close behind, while the lowest ranges were found for cefpodoxime (2/43, 4.7%) and gentamicin (1/43, 2.3%). None of the isolates showed resistance against colistin. Figure 4 depicts the occurrence of resistance against the individual antimicrobials for both sampling areas, also indicating that a higher number of isolates was retrieved from the Baltic Sea compared to the North Sea.

Evaluating resistance to the different antimicrobial classes that were tested in the present study revealed that the isolates showed the highest resistance rates to aminoglycosides (here represented by streptomycin, gentamicin, and kanamycin; 31/43, 72.1%), followed by resistance to non-extended-spectrum β-lactams (here represented by amoxicillin/clavulanic acid, ampicillin, and cefazolin; 26/43, 60.5%). Clearly lower rates were found for tetracyclines (here represented by tetracycline; 14/43, 32.6%) and folate synthesis inhibitors (here represented by sulfonamides and trimethoprim; 13/43, 30.2%), followed by quinolones (here represented by ciprofloxacin and nalidixic acid; 10/43, 23.3%) and phenicols (here represented by chloramphenicol and florfenicol; 7/43, 16.3%). The lowest rate was found for extended-spectrum β-lactamases (here represented by cefpodoxime; 2/43, 4.7%). No resistance was detected against polymyxins (here represented by colistin).

Resistance to at least one antimicrobial agent of three or more antimicrobial classes was defined as multidrug resistance and occurred in 37.2% (16/43) of the *E. coli* isolates. Of these, six were resistant to three classes, four to four classes, four to five classes, and two to six classes of the total of eight antimicrobial classes tested. Resistance to two antimicrobial classes occurred in 18.6% (8/43) of the isolates and resistance to one antimicrobial class in 44.2% (19/43). Comparing the two areas, the rate of multidrug resistance was higher in the isolates from the Baltic Sea (13/31, 41.9%) than in those from the North Sea (3/12, 25%). Interspecies differences were also observed. The highest rate of multidrug resistance was found in isolates carried by harbour porpoises (8/10, 80%), followed by those in harbour seals (5/13, 38.5%) and ringed seals (3/17, 17.6%). None of the three isolates from grey seals were multidrug-resistant. No statistically significant association was observed between the number of samples containing multidrug-resistant *E. coli* and the sampling locations (*p* = 0.3065). However, a statistically significant difference was found between marine mammal species regarding the number of multidrug-resistant *E. coli* isolates, with harbour porpoises (8/10) carrying a significantly higher number of multidrug-resistant isolates compared to the three seal species (8/33; *p* = 0.007231). A one-to-one comparison of the species revealed a significant difference between harbour porpoises and ringed seals (*p* = 0.03934), as well as between harbour porpoises and grey seals (*p* = 0.03497), but not between harbour porpoises and harbour seals (*p* = 0.0903). The number of multidrug-resistant isolates is depicted in Figure 5 for the three species carrying them.

Depending on their phenotypic resistance, isolates were tested for a selection of corresponding resistance genes. By total numbers, the most commonly detected resistance gene was *bla*_TEM_ (*n* = 16), while *bla*_OXA-1-like_ (*n* = 2) and *bla*_SHV_ (*n* = 1) genes, also mediating resistance to non-extended spectrum β-lactams, occurred only rarely. The gene *bla*_CMY-2_, which is indicative of AmpC β-lactamase production, was detected once. Higher numbers were found for genes mediating resistance to streptomycin (*strA* (*n* = 11), *strB* (*n* = 11), and *aadA1* (*n* = 7)), for genes mediating resistance to sulfonamides (*sul1* (*n* = 6) and *sul2* (*n* = 10)) and tetracyclines (*tet*(A) (*n* = 7), *tet*(B) (*n* = 8), and *tet*(D) (*n* = 3)). More rarely found resistance genes included *dfrA1* (*n* = 4) and *dfrA5* (*n* = 3), mediating resistance to trimethoprim, *floR* (*n* = 3), *catA1* (*n* = 2); *catA2* (*n* = 1), mediating resistance to phenicols; and *qnrS* (*n* = 3), mediating resistance to quinolones.

Not all isolates with phenotypic or intermediate resistance carried one of the tested resistance genes. This was the case in 16/31 aminoglycoside-resistant isolates. For isolates showing phenotypic resistance to β-lactams, no resistance genes were detected in 7/26 isolates. Furthermore, 1/7 phenicol-resistant isolates, 1/13 sulfonamide-resistant isolates, and 7/10 quinolone-resistant isolates did not carry any of the corresponding tested genes. In all isolates resistant to trimethoprim (*n* = 7) and tetracycline (*n* = 14), a corresponding gene was detected. Fourteen of the isolates without gene detection (*n* = 32) showed only intermediate resistance. These included 10 isolates with resistance to aminoglycosides, three with resistance to β-lactams, and one with resistance to phenicols. The remaining 18 isolates without resistance gene detection showed full phenotypic resistance, including seven isolates with resistance to quinolones, six isolates with resistance to aminoglycosides, four isolates with resistance to β-lactams and one isolate with resistance to sulfonamides. The detected resistance genes and the number of genes per location are summarized in Table 1. For each isolate, the resistance genotypes are shown in Figure 3.

### 3.3. Molecular Typing

The determination of major phylogenetic groups assigned 17 of the isolates to phylogenetic group B2, 13 isolates to group B1, eight isolates to group A, and five isolates to group D. The phylogenetic groups of each isolate are listed in Figure 3. The results for all of the isolates, as well as those for the four marine mammal species, are summarized in Table 2. Comparing multidrug resistance and assignment to a phylogenetic group showed that 11 (68.8%) of the 16 multidrug-resistant isolates belonged to group B1, while groups A and D each included two (each 12.5%) and group B2 included one (6.8) multidrug-resistant isolate.

For the determination of genomic relatedness, macrorestriction analyses were performed and revealed band patterns in 42 isolates. The remaining isolate (27b) was found to be non-typeable by Xbal macrorestriction. One cluster of three isolates from harbour seals from the North Sea with ≥90% similarity was identified, including two isolates from the same sampling location and sampling day and one isolate from a different sampling location and a sampling date around 4 weeks later (isolates 9, 10 and 27a). Another cluster with ≥80% similarity included three isolates from the same harbour porpoise sample from the Baltic Sea (isolates 54a, 54b and 54d), with two isolates (54b and 54d) having identical band patterns. Additionally, two isolates from three other samples had identical band patterns (isolates 39b and 39d, isolates 48a and 48b, and isolates 49a and 49b). All the other isolates showed a high degree of heterogeneity. The dendrogram of the 42 isolates is depicted in Figure 3.

### 3.4. Health Status and Antimicrobial Resistance

The results of the performed necropsies, as well as the general data of the animals, were used to evaluate whether different parameters have an influence on the occurrence of antimicrobial-resistant *E. coli*. To achieve this, the overall health status was evaluated considering all available results, classifying the necropsied animals into three health categories: good, moderate and poor. Furthermore, the two most frequent causes of disease/mortality were determined as being (broncho)pneumonia and sepsis. Other parameters of interest included body condition as well as the sex and age class of the necropsied animals. All named parameters are listed in Table 3, along with the corresponding number of animals carrying antimicrobial-resistant *E. coli*. No statistically significant association was found between the carriage of antimicrobial-resistant isolates and the different variables (overall health status (*p* = 1), (broncho)pneumonia (*p* = 0.7557), sepsis (*p* = 0.2379), body condition (*p* = 0.3616), age class (*p* = 0.9865, and sex (*p* = 0.7579)). The main pathological findings of the 41 necropsied animals are compiled in three tables in the Appendix A, including the macroscopic findings, parasitic burden, and microbiological results of the sampled organs (Appendix A), the main histopathological findings (Appendix A), along with general information, diagnoses, and overall health status (Appendix A). Additionally, Appendix A summarizes the main pathological findings, overall health status, state of nutrition, state of decomposition as well as sex, age class and indication of whether the animal was mercy killed or found deceased.

## 4. Discussion

In the present study, antimicrobial-resistant *E. coli* isolates were detected in almost 40% of the 66 faecal and rectal swabs collected from marine mammals of the North and Baltic Seas. The occurrence of resistant *E. coli* was higher in the samples from the Baltic Sea (44.4%) compared to those from the North Sea (33.3%), but without statistical significance. Although not many studies have investigated antimicrobial resistance in marine mammals thus far, resistant bacteria have been reported in various marine mammal species, including pinnipeds and cetaceans from different areas of the world [8,83,84,85,86,87]. The occurrence of resistant *E. coli* isolates varied in the different studies, with reported figures of 10% [86], 26.9% [87], 47% [8], and 50% [83]. Studies in marine mammals, including different bacterial species, reported the presence of resistant isolates in 35% to 74% of the investigated samples [83,84,86,88]. No antimicrobial-resistant *E. coli* was obtained from the fish samples collected in this study, although resistant *E. coli* has been isolated from different marine fish species in other studies [40,44].

Multidrug resistance, defined as resistance to at least one antimicrobial of three or more antimicrobial classes [89], occurred in the present study in 37.2% of the resistant *E. coli* isolates, which is in accordance with an Irish study on harbour and grey seals that found multidrug resistance in 43.5% of the *E. coli* isolates [90]. Another study investigating Enterobacterales observed a higher prevalence, with 71% of the isolates exhibiting multidrug resistance [85]. In the present study, harbour porpoises were found to carry significantly higher numbers of multidrug-resistant *E. coli* isolates compared to the investigated seal species, while there was no significant difference between species regarding the carriage of *E. coli* isolates with resistance to at least one antimicrobial drug. One other study investigating antimicrobial resistance in harbour porpoises and harbour seals found that harbour porpoises had a significantly greater risk of carrying antimicrobial-resistant bacteria compared to seals, although the same study found that all *E. coli* isolates from harbour porpoises were susceptible to the tested antimicrobials [86]. The sample size of the present study needs to be increased to verify its statistical conclusions. In addition, the comparison of species and areas might be biased due to the difference in sample sizes of the individual species for the two areas. Nevertheless, harbour porpoise samples were retrieved mainly from the Baltic Sea and harbour seal samples mainly from the North Sea, which might indicate a species difference rather than a spatial one. Furthermore, due to the delay in sample processing, it is possible that the number of *E. coli* was underestimated, although it has been shown that samples collected at larger distances and longer processing periods allow the isolation and characterization of Gram-negative bacteria [91,92,93]. Nevertheless, it cannot be excluded that the delay in sample processing could also be the reason for the observed lack of resistant *E. coli* in the sampled fishes. The reasons for the difference in the occurrence of multidrug resistance between the marine mammal species detected in the present study are unknown. The sampled harbour porpoises, harbour seals, and grey seals inhabited the same areas. Furthermore, the three seal species as well as the harbour porpoises have similar broad prey spectra consisting mainly of fish and including most of the sampled fish species [94,95,96,97]. However, one difference is that seals haul out on land in larger groups [98], while harbour porpoises predominantly occur as solitary individuals or in small groups [99]. Nevertheless, close social contacts would suggest a higher occurrence of antimicrobial resistance in seals rather than in porpoises.

Considering the tested antimicrobial classes, most studies on *E. coli* from marine mammals found the highest resistance rates against β-lactams [8,90]. In other bacteria, the most frequently reported resistances were also against β-lactams, as well as against fluoroquinolones [84,100,101]. To our knowledge, only one study on marine mammals reported high rates of resistance against aminoglycosides together with fluoroquinolones in *E. coli* [102], consistent with the finding of the present study. Here, the highest frequencies were seen for resistance to aminoglycosides (72.1%), closely followed by resistance to non-extended spectrum β-lactams (60.5%). The high rates of resistance to aminoglycosides are unexpected if comparing this result with antimicrobial drug sales in Germany, from where most of the samples were obtained. Between 2011 and 2014, the amounts of antimicrobial drugs sold to German-based veterinarians that belonged to the antimicrobial classes tested in this study were in the following order: penicillins, tetracyclines, sulfonamides, aminoglycosides, folate synthesis inhibitors, fluoroquinolones, cephalosporins, and phenicols [103]. In the same period, β-lactams and fluoroquinolones were most frequently prescribed in hospitals, while for outpatients, β-lactams and tetracyclines had the highest prescription rates [103]. This indicates that factors other than just the consumption of antimicrobial drugs in the human and animal health sectors may influence antimicrobial resistance patterns in the environment. Of specific concern is the high rate of resistance seen to streptomycin, as this is one of the watch group antibiotics in the WHO Access, Watch, Reserve (AWaRe) classification of antibiotics for the evaluation and monitoring of use [104]. On the other hand, none of the isolates showed extended-spectrum β-lactamase activity, and only one isolate was identified as being an AmpC β-lactamase producer, whereas both antimicrobial resistances were frequently detected in Enterobacterales from marine mammals of the US west coast [85,87].

In the present study, the resistance gene most frequently detected with regard to total numbers was *bla*_TEM_. Frequent occurrence of *strA*, *strB*, *aadA1*, *sul2*, *sul1*, *tet*(A), and *tet*(B) was also seen. Few instances *bla*_SHV_, *bla*_OXA-like-1_, *bla*_CMY-2_, *catA1*, *catA2*, *floR*, *tet*(D), *dfrA1*, *dfrA5*, and *qnrS* were detected. Most of these genes have already been reported in other studies of resistant bacteria from marine mammals [85,87,90]. In the present study, a corresponding resistance gene was not detectable for all resistant isolates. Eighteen phenotypically resistant and 14 intermediate resistant isolates did not carry any of the tested resistance genes. On the one hand, this can be explained by the fact that not all known resistance genes were tested, which is especially true for aminoglycosides, wherein 16 resistant isolates lacked a corresponding resistance gene [105]. On the other hand, resistances can be caused by mutations, which were also not tested here, and which are known to particularly occur in quinolone resistance, with seven of the quinolone-resistant isolates lacking a corresponding resistance gene [106,107,108]. Additionally, other resistance mechanisms such as chromosomally encoded efflux pumps may be responsible for the lack of detected resistance genes [109,110].

Resistant isolates were further assigned to the four major phylogenetic groups known for *E. coli* [81]. This assignment is primarily performed to classify isolates with regard to their phylogenesis [111]. Additionally, in *E. coli* isolates of human origin, a certain affiliation was registered regarding phylogenetic group and commensalism or pathogenicity. In this regard, isolates of phylogenetic group A are considered to be mainly commensals; isolates of group B1 are mainly commensals, but may include pathogens more often, while isolates of group D and especially group B2 are likely to be human extraintestinal pathogens [112,113,114,115]. Considering all species in the present study together, almost 40% of the isolates were assigned to the phylogenetic group B2 and then in declining frequency to groups B1 (30.2%), A (18.6%) and D (11.6%). Thus, half of the isolates are more likely to be pathogenic than commensal. Considering the individual species, harbour porpoise isolates mainly belonged to group B1, while harbour seal and ringed seal isolates most frequently belonged to group B2. The three grey seal isolates were distributed evenly between the groups A, B1 and B2. A high prevalence of *E coli* from pinnipeds in phylogenetic group B2 was also found in two other studies [116,117], while in one of these studies, another pinniped colony exhibited a higher occurrence of group D isolates [117]. This suggests a higher proportion of antimicrobial-resistant *E. coli* with pathogenic potential in seal species compared with harbour porpoises in the present study and/or hints at a different phylogenesis. In this regard, it was reported that the phylogenetic group assignment of *E. coli* isolates clearly differed in prevalence between birds (D/B1), non-human mammals (A/B1), and humans (A/B2) [118]. Another study assigned human isolates mainly to phylogenetic group A, followed by D, B2, and B1, although high variance was seen [119]. In Australia, the phylogenetic group B2 is dominant in *E. coli* isolates from humans and omnivorous terrestrial mammals, while carnivorous species mainly carry isolates of group B1 [118,120]. These classifications apply to the results presented here on harbour porpoise as non-human mammalian and piscivorous (suggesting this is grouped within carnivores) species but not for the seals, which are also non-human mammals feeding on fish. Potentially, seals are indeed carriers of group B2 isolates phylogenetically, or they may carry a high degree of human *E. coli* isolates, referring to the studies assigning human isolates predominantly to group B2. However, further research is needed to address this question. Concerning multidrug resistance, the highest frequency was found for isolates assigned to phylogenetic group B1. However, due to the difference in phylogenetic group abundance of the different species, as well as the fact that harbour porpoises carried most of the multidrug-resistant isolates, this result should be judged with caution.

In the present study, necropsy results did not reveal any statistically significant parameter that might have had an influence on the occurrence of antimicrobial-resistant *E. coli* in the investigated marine mammals. This is not surprising as there are no known negative impacts of antimicrobial-resistant bacteria on their carrier unless they are pathogenic and cause diseases that need antimicrobial treatment. Moreover, as the majority of diseased wildlife will not receive antimicrobial treatment, a treatment-associated increase in the amount of antimicrobial-resistant bacteria is unlikely to occur in diseased animals. Although our results did not show any correlation between the occurrence of antimicrobial resistance and the overall health status of wild marine mammals, elsewhere, a higher prevalence of antimicrobial resistance was reported for in stranded or bycaught marine mammals than in live marine mammals [83]. The higher prevalence of antimicrobial resistance was postulated to be potentially caused by the poorer overall health status of the stranded or by-caught animals, but could also be caused by the different sampling methods of live versus dead animals (rectal swabs versus organ samples) [83].

Finally, it should be noted that the marine mammals of the North and Baltic Seas perform extended home ranges [59,61,64,65], giving them the opportunity to disseminate antimicrobial-resistant bacteria over longer distances. The detected rates of antimicrobial resistance indicate that dissemination of the resistant bacteria into the sea as well as onto beaches may pose a direct health risk for humans [36,56], especially as humans, wildlife and the environment share similar *E. coli* strains [55,121]. This highlights the importance of monitoring sentinel species from a One Health perspective.

## 5. Conclusions

The results presented herein underscore the potential of marine mammals as sentinels for aquatic ecosystem health and human health by indicating the prevalence of antimicrobial resistance circulating in the marine environment. In addition, it is shown that marine mammals of the North and Baltic Seas serve as reservoirs and vectors for antimicrobial-resistant bacteria, thus participating in the circulation of resistant bacteria and resistance genes. The occurrence of multidrug-resistant isolates is of specific concern for human and domestic animal health. To combat antimicrobial resistance, the surveillance of different ecosystems by monitoring important reservoir and vector species is essential. Till date, there is insufficient data regarding the role of wildlife and the environment in the complex mechanism of antimicrobial resistance. Thus, it is necessary to unravel the processes underlying its emergence, preservation, and dissemination in environmental settings to ensure the effectiveness of mitigation strategies against this global health threat.

## Figures and Tables

**Figure 1 antibiotics-11-01248-f001:**
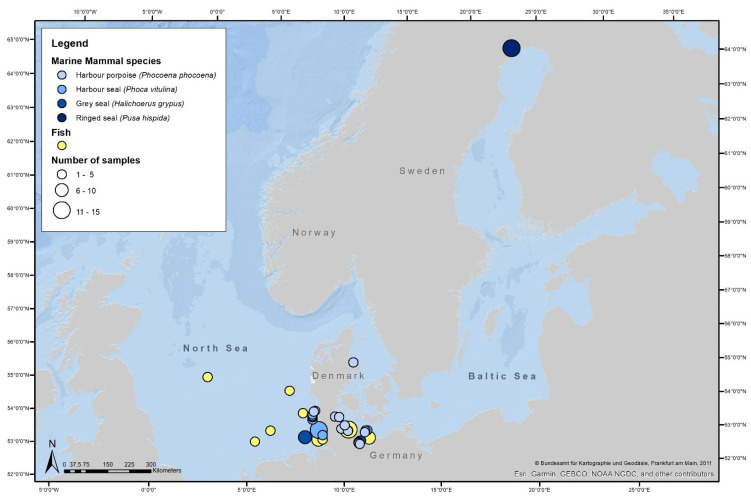
Sample map. The map indicates the sampling locations for the four marine mammal species and the fishes, as well as the sample size per location.

**Figure 2 antibiotics-11-01248-f002:**
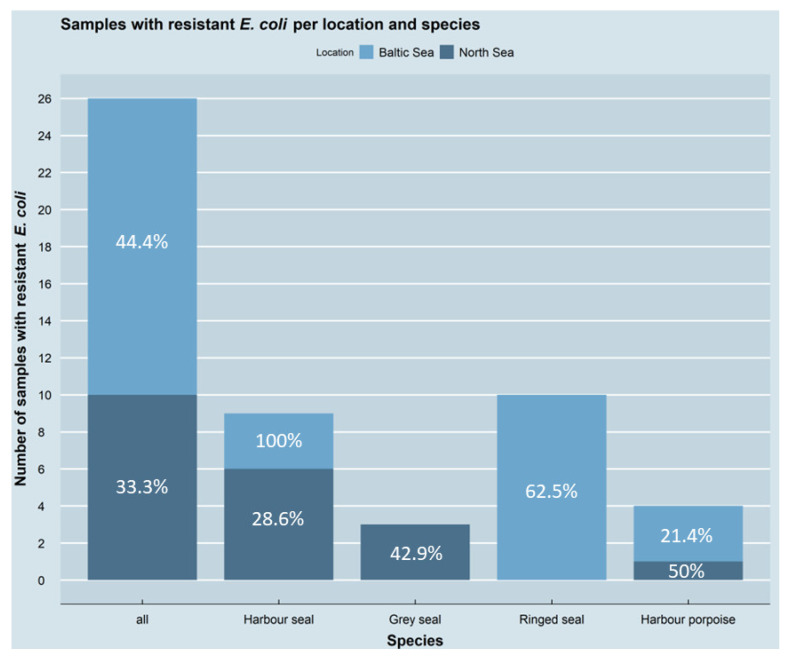
Number and percentage of resistant *E. coli* per location and species. Shown are all samples (*n* = 26) with at least one resistant *E. coli* isolate as well as their distribution over the two sampling locations and the four species. Percentages indicate proportion of samples with resistant *E. coli* isolates to the total number of samples for both locations as well as for all tested species. The figure was created using RStudio version 1.4.1103 (R version 4.1.2) with the package ggplot2.

**Figure 3 antibiotics-11-01248-f003:**
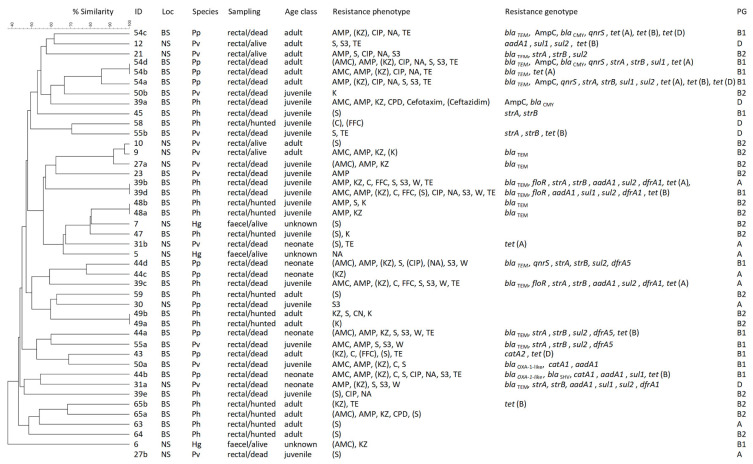
Overview of the pheno- and genotypes of *E. coli* isolates from free-ranging marine mammals. The dendrogram is shown containing 42 isolates; one isolate was not typeable. For all isolates, the sampling location, the species, the sampling method, the age class, the resistance pheno- and genotypes, and the phylogenetic group are listed. Locations: NS = North Sea, BS = Baltic Sea. Species: Pp = harbour porpoise, Pv = harbour seal, Ph = ringed seal, Hg = grey seal. Resistance phenotype: AMC = amoxicillin/clavulanic acid, AMP = ampicillin, KZ = cefazolin, CPD = cefpodoxime, C = chloramphenicol, FFC = florfenicol, S = streptomycin, CN = gentamicin, K = kanamycin, CIP = ciprofloxacin, NA = nalidixic acid, S3 = compound sulfonamide, W = trimethoprim, TE = tetracycline. Antimicrobials in brackets indicate intermediate resistance.

**Figure 4 antibiotics-11-01248-f004:**
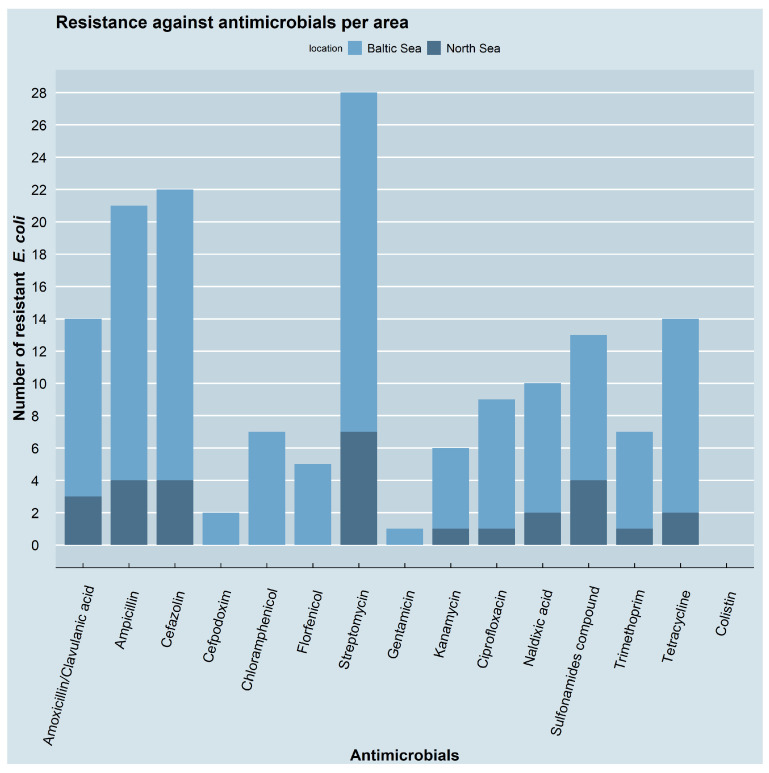
Resistance against antimicrobials by location. The number of resistant *E. coli* isolates is shown for each of the tested antimicrobials. The proportion of the two locations is indicated by different colours with blue for the Baltic Sea and dark blue for the North Sea. Isolates with more than one resistance are included multiple times. The figure was created using RStudio version 1.4.1103 (R version 4.1.2) with the package ggplot2.

**Figure 5 antibiotics-11-01248-f005:**
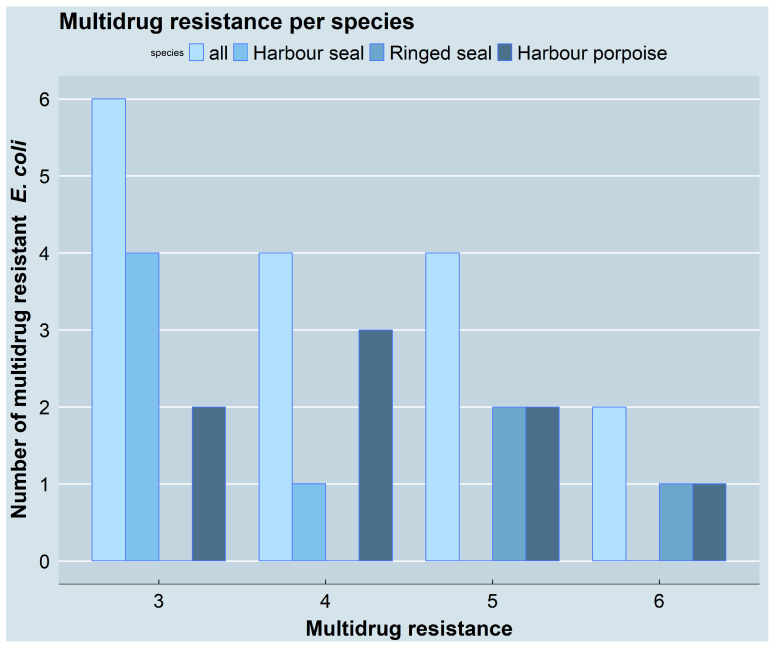
Multidrug resistance. The graphic depicts the number of multidrug resistant isolates with resistance against 3, 4, 5, and 6 antimicrobial classes for all species as well as separated for the individual species. The figure was created using RStudio version 1.4.1103 (R version 4.1.2) with the package ggplot2.

**Table 1 antibiotics-11-01248-t001:** Detected resistance genes. Total number of resistance genes as well as number of resistance genes per species. In the three isolates of grey seals no resistance gene was detected.

Resistance Gene	All	Harbour Seal	Ringed Seal	Harbour Porpoise
*bla* _TEM_	16	5	5	6
*bla* _SHV_	1	0	0	1
*bla* _OXA-1-like_	2	1	0	1
*bla* _CMY_	1	0	1	0
*strA*	11	4	3	4
*strB*	11	4	3	4
*aadA1*	7	3	3	1
*catA1*	2	1	0	1
*catA2*	1	0	0	1
*floR*	3	0	3	0
*tet*(A)	7	1	2	4
*tet*(B)	8	2	2	4
*tet*(D)	3	0	0	3
*sul1*	6	2	1	3
*sul2*	10	4	3	3
*dfrA1*	4	1	3	0
*dfrA5*	3	1	0	2
*qnrS*	3	0	0	3

**Table 2 antibiotics-11-01248-t002:** Phylogenetic groups. Number and percentage of phylogenetic groups of isolates from the four sampled marine mammal species, as well as for all the isolates.

Phylogenetic Group	A	B1	B2	D
All	8 (18.6%)	13 (30.2%)	17 (39.5%)	5 (11.6%)
Harbour seal	2 (15.4%)	2 (15.4%)	6 (46.2%)	3 (23.1%)
Grey seal	1 (33.3%)	1 (33.3%)	1 (33.3%)	0
Ringed seal	3 (17.6%)	2 (11.8%)	10 (58.8%)	2 (11.8%)
Harbour porpoise	2 (20.0%)	8 (80.0%)	0	0

**Table 3 antibiotics-11-01248-t003:** Antimicrobial resistance as a function of health parameters, sex and age class. Occurrence of antimicrobial resistance listed for different parameters including overall health, body condition, sex, age and the two most frequent diseases diagnosed during marine mammal necropsies (*n* = 41) with 20 animals harbouring antimicrobial-resistant *E. coli*.

Antimicrobial-Resistant Isolates		Yes	No
overall health	good	2	3
	moderate	10	11
	poor	8	7
body condition	good	6	12
	moderate	6	5
	poor	4	3
sex	male	11	10
	female	9	11
age	neonate	2	1
	juvenile	11	13
	adult	7	6
(broncho)pneumonia	yes	6	6
	no	14	15
sepsis	yes	8	10
	no	12	11

## Data Availability

The data presented in this study are available in this published article and its Appendix A.

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
