# Peer review of "Occurrence of Antimicrobial-Resistant Escherichia coli in Marine Mammals of the North and Baltic Seas: Sentinels for Human Health"

_antibiotics, 2022, doi:10.3390/antibiotics11091248_

Round 1
Reviewer 1 Report
Comments on Article: antibiotics-1908245
Title: Occurrence of antimicrobial-resistant Escherichia coli in marine mammals of the North and Baltic Seas sentinels for human health
General Comments:
The Article describes the study on occurrence of antimicrobial-resistant Escherichia coli in 66 marine mammals and 40 various fish species of the North and Baltic Seas. According to the Results, the authors have isolated 39.4% resistant E. coli (37, 2% multidrug resistant strains) from marine mammal samples and no resistant strains in samples from fish. Isolated strains were assigned to the four major phylogenetic groups of E. coli. The Results of this study are interesting and show distribution of antimicrobial-resistant E. coli in four different marine mammal species.
The article provides a lot of data in this research area and all the sections are well written. There are only minor questions regarding the article at specified bellow. After addressing these questions, I recommend it for publishing in your Journal.
Specific Comments:
Page 3, Lines 140-165. The procedure with swabs after sampling should include more detail. Did the authors wetted the swabs and used some transfer solution, or?
Page 5, Lines 193-194. Please rewrite this sentence to make it clearer.
Page 5, Lines 206-207. More details should be provided. Stored as dry swabs or in some solution? Did the authors question the survival of microorganisms in this kind of storage? What about potential competitive microflora? Moreover, did they use any pre-enrichment broth to improve recovery of (pathogenic) bacteria?
Are there any references for this method for isolation of bacteria? Did the authors use quality check (positive and negative control)? What would be the survival rate of bacteria on swabs that are "stored at room temperature between one week and almost nine months"?
Page 6, Lines 264-267. Did the authors test the true number of bacteria (in 0.5 MacFalrdand)?
Page 16, Lines 612-615. Sentences that explain/comment the higher prevalence in should be added here.
Reviewer 2 Report
This paper entitled"Occurrence of antimicrobial-resistant Escherichia coli in marine
mammals of the North and Baltic Seas: sentinels for human health" is appeared to be a nice piece of work and will provide more information and reference for future study. However, I think this need minor revision to accept. There are some small problems that need attention, such as, line 326, Nine of 24 harbour seal samples, the arabic numerals are used before and after the article, and this needs to be unified. line 334, 335, E. coli need Italics in the paper.
